# Genetic and Pathogenic Variability of *Mycogone perniciosa* Isolates Causing Wet Bubble Disease on *Agaricus bisporus* in China

**DOI:** 10.3390/pathogens8040179

**Published:** 2019-10-08

**Authors:** Dan Li, Frederick Leo Sossah, Yang Yang, Zhenghui Liu, Yueting Dai, Bing Song, Yongping Fu, Yu Li

**Affiliations:** 1Engineering Research Center of Chinese Ministry of Education for Edible and Medicinal Fungi, Jilin Agricultural University, Changchun 130118, China; 2School of Plant Protection, Jilin Agricultural University, Changchun 130118, China

**Keywords:** AFLP, *Mycogone perniciosa*, wet bubble disease, pathogenicity, diversity

## Abstract

Wet bubble disease, caused by *Mycogone perniciosa*, is a major threat to *Agaricus bisporus* production in China. In order to understand the variability in genetic, pathogenicity, morphology, and symptom production of the fungus, 18 isolates of the pathogen were collected from diseased *A*. *bisporus* in different provinces in China. The isolates were characterized by a combination of morphological, cultural, molecular, and pathogenicity testing on different strains of *A*. *bisporus* and amplified fragment length polymorphism (AFLP) analysis. The 18 isolates were identified by Koch’s postulate and confirmed different pathogenic variability among them. The yellow to brown isolates were more virulent than the white isolates. AFLP markers clustered the isolates into two distinct groups based on their colony color, with a high level of polymorphism of Jaccard similarities range from 0.39% to 0.64%. However, there was no evidence of an association between the genetic diversity and the geographical origin of the isolates. Through knowledge of the genetic diversity, phenotypic virulence of *M. perniciosa* is a key factor for successful breeding of resistant strains of *A*. *bisporus* and developing of an integrated disease management strategy to manage wet bubble disease of *A*. *bisporus*.

## 1. Introduction

*Agaricus bisporus* (Lange) Imbach, the button mushroom, is an edible and medicinal basidiomycete that belongs to the family Agaricaceae. It is widely distributed, and one of the most cultivated and consumed mushrooms in the world. China is one of the largest producers of *A. bisporus* in the world [1]. The global demand for healthy and nutritious food has led to the continuous expansion of the cultivation area by indigenous farmers and industrial factories in China [2]. However, the production of *A*. *bisporus* is severely constrained by diseases such as *Mycogone perniciosa*, the causal agent of wet bubble disease [3]. Wet bubble disease causes significant economic losses in button mushroom worldwide [4,5]. In China, yield losses of about 15–30% are due to the occurrence of this disease, while rapid epidemics or severe infections can lead to total mushroom loss [6]. 

*Mycogone perniciosa* can cause a wide range of symptom severity during infection and the disease is usually controlled by cultural practices coupled with the application of fungicides [3]. However, none of the present control measures have been found completely effective due to resistance development to fungicides [3,6]. The resistance of some commercial and wild strains of *A*. *bisporus* to *M. perniciosa* is known [7]. However, genes governing resistance have not yet been identified. In addition, there is variability in the morphological traits (appearance, growth rate, colony color), and pathogenicity of *M. perniciosa* isolates [4]. 

During the past few decades, the molecular methods widely used to evaluate the genetic diversity among *M. perniciosa* isolates include restriction fragment length polymorphism (RFLP) [4], random amplified polymorphic DNA (RAPD) [7], inter-simple sequence repeats (ISSR) [8], sequence-related amplification polymorphism (SRAP) [9], and sequence-characterized amplified region (SCAR) markers [10]. Compared to the above molecular methods, the amplified fragment length polymorphism (AFLP) provides a higher resolution, higher efficiency in detecting polymorphism, more reliable, reproducible, and easily transferable to other markers such as simple sequence repeats (SSR) [11]. Hence, it is mainly applied to genetic variability and population biology studies. Previous work by several authors [7,8,9,10] using RFLP, SRAP, ISSR, and SCAR markers shows the existence of high genetic variability in *M. perniciosa* isolates. Despite the severe economic losses caused by wet bubble disease in China, there is limited information about the genetic and pathogenic diversity of *M. perniciosa* isolates in the country and whether the wide variation in virulence is associated with the phenotype or geographical origin of the isolates. 

The aim of the present study was to (1) characterize 18 isolates of *M. perniciosa* producing wet bubble disease symptoms on *A. bisporus* in four different provinces in China using morphological and cultural characters and phylogenetic analysis of the internal transcribed spacer (ITS) region of the ribosomal DNA, and (2) assess the pathogenic variability on six *A. bisporus* strains and estimated the genetic diversity using AFLP markers. 

## 2. Results

### 2.1. Morphological Characterization and Phylogenetic Analyses of Mycogone perniciosa

Fungal isolates were collected from infected *A*. *bisporus* basidiome showing symptoms of wet bubble disease from different provinces (Gansu, Fujian, Shandong, and Hubei) of China (Table 1). Eighteen isolates were initially identified as *M. perniciosa* on the basis of colony morphology and microscopic observations. The mycelia color of the isolates ranged from white to dark brown [4] on PDA for seven days at 25 °C (Figure 1). The colony texture was either villous or concentric circles, with all white-colored colonies having villous texture. The average colony growth rates (colonies were 7.1 to 18 mm in diameter) was 11.73 mm/day. The spores of *M. perniciosa* in the anamorph stage are divided into two types: one is the conidia with thin cell wall, ellipse, colorless and transparent, which are born on the conidia pedicel with a diameter of 9.77 × 4.25 µm–12.77 × 6.05 µm (averge, n = 60) and the other one is the chlamydospore composed of the upper and lower cells. The upper cell is yellow-brown to gray-brown, and the lower cell is colorless and transparent, its size is from 11.36 × 8.08 µm–33.43 × 25.94 µm (averge, n = 60).

On PDA chlamydospores were not observed on Hp2 and Hp9, similarly, conidia were not observed for H2, Hp8, and Hp9 (Appendix A). The isolates with brown colony morphology produced more conidia compared to the isolates with white colony morphology. The cultural and morphological characteristics of all isolates were consistent with those of the genus *M. perniciosa* (Magn.) Delacroix (teleomorph = *Hypomyces perniciosus* Magnus) [4]. The sequencing of the amplified PCR product of the ITS gene for the isolates resulted in an amplicon size of 750 bp. Blast search of the sequences of the individual isolates revealed a 99% homology to *M. perniciosa* sequences at the NCBI GenBank database. All of the 18 isolates of *M. perniciosa* were pathogenic on *A. bisporus* strain CCMJ1020 resulting in brown spot, malformation, and water-soaking symptoms typical for wet bubble disease. The *M. perniciosa* inoculated button mushrooms showed symptoms similar to those observed in the field (Figure 2) and the fungi were consistently re-isolated but not from controls, thus fulfilling Koch’s postulate, while the controls exhibited no symptoms. Based on the colony morphology, cultural characteristics, and phylogenetic analysis [9], and the pathogen was identified as *M. perniciosa* (Magn.) Delacroix. All resulting sequences were deposited in NCBI GenBank; accession numbers are shown in Table 1.

### 2.2. Pathogenicity Variability

The *A. bisporus* strains expressed wet bubble disease symptoms 2–5 days post-inoculation of *M. perniciosa* isolates using a concentration of 1 × 10^5^ conidial suspension. The minimum incubation period was 2–3 days for all isolates on the highly susceptible *A. bisporus* strains and those for highly resistant *A. bisporus* strains required a maximum incubation period of five days. The wet bubble disease symptoms on *A. bisporus* include malformation of basidiome, white, fluffy mycelial growth, copious amber droplets, and flocculent mycelia on most substrates [12]. All the isolates produced typical wet bubble disease symptoms with different levels of symptom severity depending on all the *A. bisporus* strains tested. There was a constant increase in the symptom severity scores from seven to 30 days post-inoculation during disease progression. Out of the 18 isolates caused disease on all the tested *A*. *bisporus* cultivars and had a severity score of more than 50%. Seven did not produce any visible symptoms on *A. bisporus* strain CCMJ1110. The most virulent isolate was Hp10 collected from Hubei province. Most of the isolates with lower virulence levels were those with white colony morphology. The least virulent isolate was Hp1 collected from Gansu province (see Table 2).

### 2.3. AFLP Analysis

The *M. perniciosa* AFLP fingerprints produced a total of 1221 scorable loci from the combination of 10 primer pairs. The number of bands generated per isolate ranged from a minimum of 108 bands by primer D-2 to a maximum of 130 bands by primer pairs B-1 and B-8. The analysis revealed a total of 1206 (98.8%) polymorphic AFLP fragments ranging in size from 69.5 bp to 499.7 bp. The AFLP experiment was repeated three times and all the polymorphic DNA fragments were identical. The dendrogram (Figure 3) for the 18 isolates were drawn from the genetic similarity data using the UPGMA method with the Jaccard’s pairwise similarity coefficients. The isolates clustered into two well-defined distinct groups with a mean similarity coefficient of 0.39. The isolates were not clustered based on their geographic location. Cluster A (*s* = 0.495–0.6) grouped five isolates, two collected from Gansu (Hp1 and Hp2) and three from Fujian (Hp3–Hp5) provinces of China. The isolates from cluster A were all pathogenic to the susceptible *A*. *bisporus* strains (CCMJ1020). None of the isolates in this cluster was pathogenic on the resistant *A*. *bisporus* (CCMJ1106 and CCMJ1110) strains. In addition, they caused mild forms of wet bubble disease on *A. bisporus* strains. Thirteen isolates were grouped into cluster B (*s* = 0.46–0.64) and they were collected from Fujian, Gansu, Shandong, and Hubei. The isolates in cluster B were yellow to brown and they caused the severe forms of wet bubble disease on *A. bisporus*. Within the clusters, the isolates showed a high degree of variability, however, isolates Hp11 and Hp 12 in cluster B showed the greatest similarity at 0.64 (Figure 3 and Figure 4). 

## 3. Discussion

*Mycogone perniciosa* is a recurrent pathogen causing substantial yield losses on diverse edible mushrooms (*A. bisporus*, *Pleurotus ostreatus, P. citrinopileatus, Volvariella volvacea*, etc.) [4,13] during severe epidemics globally. In this study, we investigated the genetic variability and pathogenicity (virulence) of *M. perniciosa* from major *Agaricus bisporus* growing areas in China. Eighteen isolates of *M. perniciosa* causing wet bubble disease on *A*. *bisporus* in Fujian, Hubei, Gansu, and Shandong were identified based on their colony morpho-cultural characters and confirmed by molecular characterization using ITS region and pathogenicity testing on *A. bisporus* strain CCMJ1020 [3,7,14]. The subunits of ribosomal DNA (rDNA) genes have been widely used in fungal identification [15]. Consequently, the results indicate the importance of a combination of molecular and morphological data for unambiguous identification of fungal species [16]. The inoculation of *M. perniciosa* isolates on the six different *A*. *bisporus* strains, the consequential incubation period and wet bubble disease development showed the variation in their pathogenicity and virulence. The *M. perniciosa* isolates were low, moderate, and highly virulent based on their symptom development and the number of different *A*. *bisporus*. The white, yellow, and brown white isolates produced mild, moderate, and severe symptoms with corresponding increasing disease severity scores. The brown isolate (Hp 10) had the highest virulence level, which caused disease on all the different strains of *A*. *bisporus*. The variability in virulence of isolates indicates that there must be different pathotypes in *M. perniciosa* populations, therefore, it requires proper pathogenicity test on different strains of *A*. *bisporus* to clearly distinguish the pathogenic variability. 

Isolates (Hp2, Hp8, and Hp9) of *M. perniciosa* did not produce conidia, so the mycelial suspension was used as inoculum for pathogenicity testing. From our analysis, we could not determine the extent to which this type of primary inoculum might have influenced the results of the virulence level of the isolates [17]. Fletcher et al. [4] also used symptom severity to characterize the virulence of *M. perniciosa*. The severity of disease symptoms is dependent on the isolate, spore inoculum and the host tolerance levels to the pathogen [18]. The white isolates of *M*. *perniciosa* showed less pathogenicity compared to the yellow/brown isolates. This may be due to the colorless conidia and a lower degree of sporulation compared to the yellow/brown colonies. For example, Fletcher et al. [4] reported that non-pigmented white isolates of *M*. *perniciosa* are less pathogenic than the pigmented isolates. The moderate to high virulence levels of the yellow to brown *M. perniciosa* isolates may be due to a large amount of conidia production, and production of secondary metabolites (e.g., pigments and mycotoxins) plays an important role in pathogenesis [12,19]. In contrast, Fletcher et al. [4] reported that the less aggressiveness of some of the isolates maybe as a result of mycovirus infecting those strains. However, in this study, we did not test the presence of virus-like particles in any of the isolates. In addition, coevolution of a pathogen and host, host adaptation or response to the pathogen may lead to variation in virulence or virulent pathotypes [20,21,22]. 

AFLP analysis was used to infer the genetic variability of the 18 *M. perniciosa* isolates. The results reveal high genetic diversity among the isolates. The genetic diversity observed using AFLP is higher than those conducted in China using SRAP, and RAPD, ISSR, SRAP, and SCAR marker respectively [7,8,9,10]. Although it is difficult to compare levels of genetic diversity obtained with different markers and different sample sizes [23]. The AFLP technique has been shown to be a very powerful method for genetic diversity studies, because it detects many polymorphisms in a single assay, shows reasonable coverage of the genome, and is highly repeatable [24]. The application of AFLP fingerprinting was sensitive enough to distinguish between the isolates, thus provides a powerful tool for distinguishing inter- and intra-species [25]. The variability in the morphological and cultural characteristics of the isolates was supported by the AFLP analysis. The isolates were clustered into two groups based on their morphology; however, there was no correlation between the clustering and their geographical origin. This may be due to the large diversity of the pathogen population in China. Variations in cultural characteristics have been observed in the isolates of *M. perniciosa* [4]). The AFLP data showed that Hp11 and Hp 12 were genetically similar (Figure 4). Though both isolates were collected from Shandong province, they were isolated from different cities. This could be due to the exchange of *A*. *bisporus* material from different locations. 

Mutation, gene flow, sexual reproduction (recombination) and the adaptation to diverse fungal hosts might have contributed to the genetic diversity of *M. perniciosa* from the AFLP analysis. In our previous work [12], the analysis of the genome sequence of *M. perniciosa* (Hp 10) revealed it is heterothallic species containing *MAT 1-2* locus. There is a need for further studies to identify the sexual stages of the isolates to find out whether the sexual stage contributes to the diversity of this pathogen [26]. In addition, the diversity of host germplasm, cultivation method of *A*. *bisporus*, or environmental factors may play a role in determining the diversity of the pathogen. 

## 4. Materials and Methods 

### 4.1. Fungal Isolates and Morphological Characteristics 

*Mycogone perniciosa* isolates (Table 1) were obtained from the fruiting bodies of *A. bisporus* showing typical symptoms of wet bubbled disease in mushroom farms located in Fujian, Gansu, Hubei, and Shandong provinces of China. The disease survey was carried out in 2014–2015. The sampled infected tissues were sterilized in 2% sodium hypochlorite (NaClO) solution for 60 secs and washed three times with sterilized deionized (DI) water, then plated on Petri dishes containing PDA (potatoes 200 g/L, glucose 20 g/L, agar 15 g/L) amended with 100mg/L Kanamycin and incubated at 25 °C for 5 days in darkness. Pure cultures were subsequently obtained through single spore isolation from all colonies showing the morphological characteristic of a typical *M. perniciosa* and five representative purified isolates sub-cultured on PDA without antibiotics. The growth characteristics, colony morphology, and conidial characteristics—such as shape, length, width—were examined for a total of 18 representative isolates. Colony color was assessed 7–10 days after single spores were transferred to PDA. A minimum of 30 conidial characters was observed under a Leica DMR HC microscope (Leica Microsystems Imaging Solutions Ltd., Cambridge, UK) fitted with Leica DFC320. The sporulation of the isolates was estimated as described by Santos et al. [27]. Briefly, 100 mg of the fungal mycelium of each isolate was collected and transferred to an Eppendorf tube, in which it was homogenized with 1 mL of Tween 80 solution (0.05%). The conidia count of each such suspension was then determined using a Neubauer chamber. Conidia counts were performed in triplicate for each isolate. All cultures were conserved on PDA in slant tubes and deposited in the Engineering Research Center of the Chinese Ministry of Education for Edible and Medicinal Fungi of Jilin Agricultural University (HMJAU) in China.

### 4.2. DNA Extraction and Molecular Identification

Total genomic DNA was extracted from 7-day old mycelia mat growing on PDA plates with cellophane sheets using the Nuclear Plant Genomic DNA Kit of CWBIO (CWBIOTECH, Beijing) following the manufacturer’s protocol. The DNA quality and quantity were measured using a BioSpec-nano spectrophotometer (Shimadzu Biotech, Tokyo, Japan) at a wavelength of 260 and 280 nm, respectively. The DNA was stored at −80 °C until required for further use.

PCR amplification and sequencing of the internal transcribed spacer regions of the rDNA was performed for each isolate utilizing the primer set ITS4 and ITS5 [28]. The obtained sequences were individually checked by BLAST analysis against the NCBI GenBank (http://www.ncbi.nlm.nih.gov/) database and highly corresponding sequences were retrieved, aligned, and the phylogenetic tree constructed with the maximum likelihood method using the Tamura and Nei substitution method [29] in MEGAX [30].

### 4.3. Pathogenicity Tests

#### 4.3.1. Mushroom Strains and Cultivation Method

Pathogenicity and virulence were carried out at the Mushroom Base of Jilin Agricultural University, Changchun, China from September 2015 to March 2017. The pathogens were tested on six different strains of *A. bisporus* with known different level of disease resistance [7] to *M. perniciosa* (highly susceptible (CCMJ1020, CCMJ1036) moderately resistant (CCMJ1053, CCMJ1074) and highly resistant (CCMJ1106, CCMJ1110) collected from the Herbarium of the Institute of Mycological, Jilin Agricultural University, China). The *A. bisporus* strains were grown in baskets with dimensions (45 × 33 × 25cm) filled with 7.5 kg of compost. About 4 cm thick casing soil was applied to cover the compost when the mycelium overgrew the compost. To induce fruiting, the room temperature was set at 21–24 °C, relative humidity control to 80–95%, and the CO_2_ concentration was in 1200–1500 ppm.

#### 4.3.2. Inoculum Preparation and Disease Assessment

The inoculum for each *M. perniciosa* isolates was prepared from 7-day old cultures on PDA, by washing down the pathogen conidia with sterile distilled water and sieving the deferment via six layers of sterile cheesecloth. The spore/conidial concentration was estimated using a hemocytometer. The optimal spore/conidial concentration of the suspension for the isolates to cause disease was determined by inoculating 10^4^, 10^5^, 5 × 10^5^, and 10^6^ conidia/ml suspension on *A. bisporus* strains CCMJ1020 and CCMJ1036. A spore/conidial concentration of 1 x 10^5^ was standardized and used for disease inoculation for all the *A. bisporus* strains. 

Three days after the application of casing soil and regulation of temperature and relative humidity, approximately 50 ml of inoculum (spore/ conidial concentration10^5^/ml) were sprayed into each basket containing the cultivated button mushroom. The controls were sprayed with 50 ml of sterile distilled water. Three replications were evaluated per isolate per mushroom strain. Disease assessment was recorded for the first flush. Pathogenicity was determined by the number of *A. bisporus* strains on which an isolate caused wet bubble disease and the order of susceptibility of strains to individual isolates. Pathogenicity tests to confirm Koch’s postulate was assessed on *A. bisporus* strain CCMJ1020. Disease severity was rated on sporocarp of individual mushroom strains for 30 days after inoculation using a 0–5 visual rating scale, where 0 = no symptom; 1 = 1–10%; 2 = 11–25%; 3 = 26–50%; 4 = 51–75%; and 5 = >75% based on the number of sporocarp showing disease against the total mushrooms harvested from the baskets. Based on the rating scale, the *A. bisporus* strains were classified as either resistant or susceptible (≤ 3 = resistance (R) and >3 = susceptible (S)). The severity indexes were subjected to one-way analysis of variance, and significant mean differences (*P* = 0.05) were determined with Duncan’s multiple range test using GenStat 12th Edition version 12.0.0.3033 (VSNI, Hemel Hempstead, England). The experiment was repeated three times in a completely randomized design with three replicates per *M. perniciosa* isolate. The same batch of compost was used for each experimental trial. Also prior to each trial, the pathogenicity of each isolate was tested on the *A. bisporus* caps to confirm their pathogenicity before the trial. All trials were subjected to the same environmental conditions (temperature and relative humidity) and routine rigid management was maintained in a clean environment to prevent contamination from other pathogens. *M. perniciosa* isolate WH001 inoculated on *A. bisporus* strain CCMJ1020 was used as a standard for each trial to detect the effect of variation in growth room conditions on symptom expression. 

### 4.4. AFLP Analysis

The AFLP reactions were carried out as described by Vos et al. (1995) [31] with modifications. The adapters and primers used in this study are shown in Table 3 and were purchased from Genset Oligos, France and IBB PAN, Poland. Restriction digestion and adapter ligation were performed simultaneously in a 20 μL reaction volume made of 4 μL (50 ng/μL genomic DNA, 1 μL Adapter, 2 μL (5 units (U)) HindIII/MseI (New England Biolabs Inc., Ipswich, MA, USA), 2.5 μL 10X Reaction buffer, 2.5 μL 10 mM ATP, 1 μL (1 unit) T4 DNA Ligase (New England Biolabs Inc., Ipswich, MA, USA) and 7 μL H_2_O. The reaction mixture was centrifuged for 15 s, incubated at 37 °C for 5 h, held at 8 °C for 4 h and stored overnight at 4 °C. The quantity and quality of the digested products were observed using electrophoresis on 1.5% agarose gels stained with GelRed, visualized and photographed using Bio-Rad Gel Doc XR+ system (Bio-Rad Laboratories Inc., Hercules, CA, USA). 

Nonselective PCR pre-amplification was performed on the digested and ligated template DNA using non-selective primer pair HindIII/MseI in a total volume of 25 μL. The PCR was performed in a T-Personal thermal cycler (Biometra, Göttingen, Germany) with the following settings: 94 °C for 2 min followed by 30 cycles of 30 s at 94 °C, 30 s at 56 °C, and 80 s at 72 °C. The final thermal cycle was followed by a 5 min extension at 72 °C and (hold temperature conserved at 4 °C for the moment) and stored at −20 °C before gel electrophoresis. The PCR products were diluted 20-fold with TE buffer. The selective PCR amplification was performed in 25 µL total volume containing eight different primer pairs consisting of HindIII combined with MseI (Table 3). All reactions were carried out in a T-Personal thermal cycler (Biometra, Göttingen, Germany) with the following settings; first-round amplification, 94 °C for 2 min followed by 12 cycles of amplification, with a decreasing annealing temperature of 0.7 °C/cycle: 94 °C for 30 s, first annealing for 30 s at 65 °C (the annealing temperature was influenced by primers Tm), 72 °C for 80 s, and next 23 amplification cycles of 94 °C for 30 s, 55 °C (the annealing temperature was influenced by primers Tm) for 30 s, and 72 °C for 80 s. The final thermal cycle was followed by the extension of 5 min at 72 °C. The PCR yields were stored at 4 °C till subsequent analysis. Five µL of loading buffer (GelT^M^ Vilber Lourmat, Collégien, France) were added to 25 µL of the PCR products. The mixture was loaded on 1% polyacrylamide gel in 1 × TBE buffer (89 mM boric acid, 89 mM Tris base, 2 mM EDTA, pH8.0) and run in the Agagel Mini, Biometra electrophoresis system (Biometra, Göttingen, Germany) was run at 200 V in TBE buffer for 20 min. The gels were stained with GelRed (Biotium, Inc., Fremont, CA, USA), visualized and imaged on a UV transilluminator (Vilber Lourmat FLX-20M, Collégien, France). The DNA samples of each isolate were extracted three times from fresh fungal cultures and fingerprinted twice to estimate the reproducibility of the AFLP band patterns. The electrophoretograms were examined using GeneScan^®^ Analysis Software (Applied Biosystems, Inc., Foster City, CA, USA). AFLP markers were physically scored as binary data for the existence or nonexistence of fragments between 70 and 500 bp. This binary data obtained was later used to estimate the Jaccard’s pairwise similarity coefficients as applied in the FreeTree version 0.9.1.50 program [32]. The unweighted-pair-grouping method with arithmetic average (UPGMA) dendrogram was produced from DNA band patterns using the Nei and Li correlation coefficient [14]. The phylogenetic tree was viewed and edited using NTSYSpc version 2.02 (Exeter Software, Setauket, New York, USA).

## 5. Conclusions

This study showed the variation in virulence and demonstrated the applicability of the AFLP technique to distinguish the genetic variation among *M. perniciosa* isolates from China. However, there is a need for continuous monitoring of mushroom farms in the various geographical areas in China to collect a large sample population and also compare them with other isolates from different countries for a more detailed genetic analysis, population structure, and evolution of *M. perniciosa*. Furthermore, identification and development of genetic markers associated with pathogenicity (virulence) will aid in selecting sources of disease resistance and breed *A*. *bisporus* with broad resistance to the various pathotypes of *M. perniciosa*, in order to develop a more effective management strategy to control wet bubble disease in China. 

## Figures and Tables

**Figure 1 pathogens-08-00179-f001:**
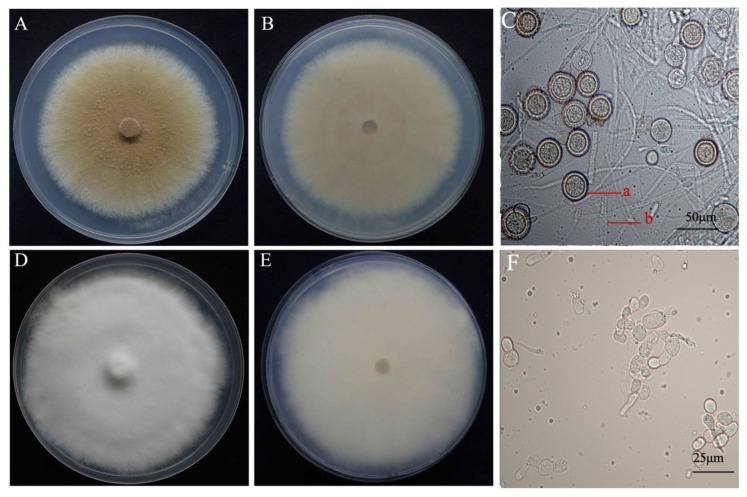
Colony morphology and microscopic characteristics of *Mycogone perniciosa*: colony color. (**A**) upper, (**B**) under, (**C**) microstructure for isolate HP10. Colony color (**D**) upper, (**E**) under, (**F**) microstructure for isolate Hp2. (**a**) aleuriospore; (**b**) hypha.

**Figure 2 pathogens-08-00179-f002:**
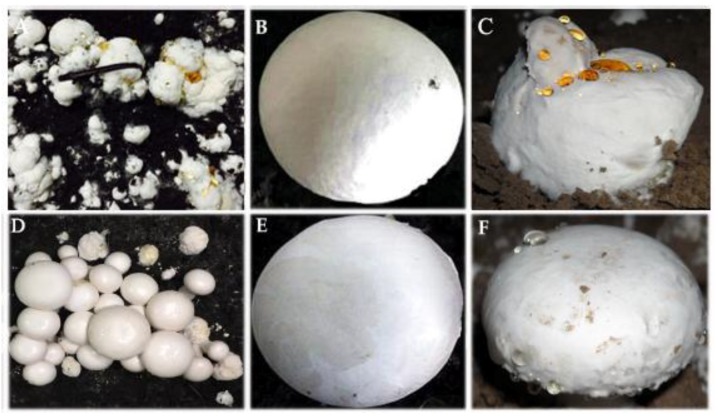
Pathogenicity tests for *Mycogone perniciosa* on *Agaricus bisporus* strain CCMJ1020: isolate Hp10. (**A**) Diseased basidiome occurring in the field, (**B**) Control, (**C**) wet bubble disease from pathogenicity test. Isolate Hp2 (**D**) Diseased basidiome occurring in the field, (**E**) control, (**F**) wet bubble disease from pathogenicity test.

**Figure 3 pathogens-08-00179-f003:**
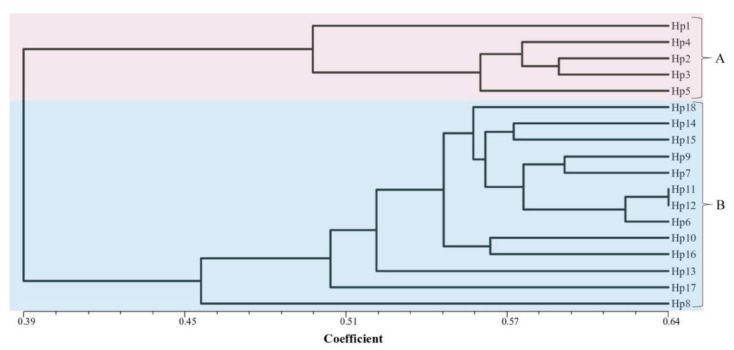
UPGMA (unweighted pair group method with arithmetic mean) dendrogram of *Mycogone perniciosa* isolates based on the genetic similarity matrix obtained with the Jaccard index for the data from AFLP markers.

**Figure 4 pathogens-08-00179-f004:**
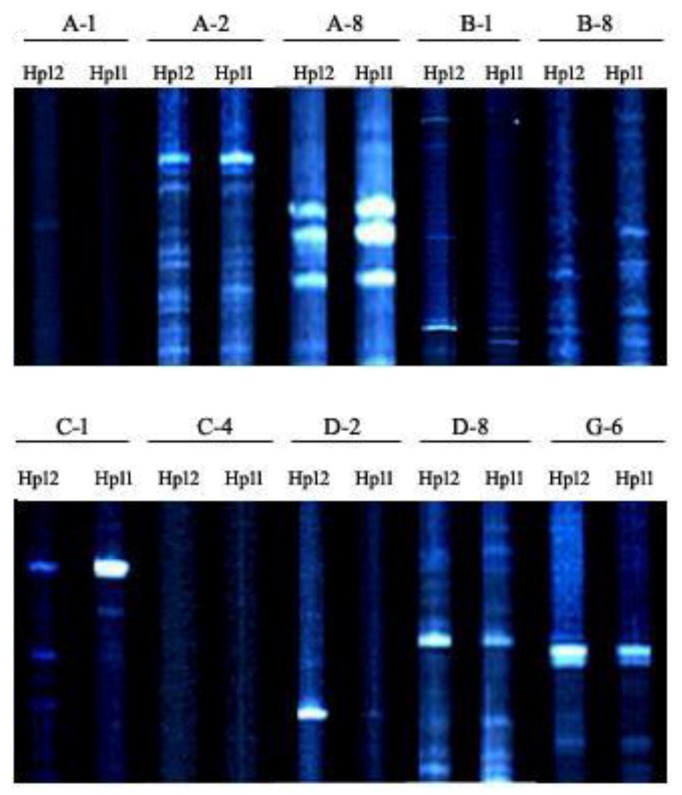
Gel image showing the AFLP banding pattern of Hp11 and Hp 12 of *Mycogone perniciosa*.

**Table 1 pathogens-08-00179-t001:** Morphological characteristics of 18 isolates of *Mycogone perniciosa* growing on PDA medium at 25°C for 7 days.

No	Isolates	Origin	Colony			Conidia								GenBank Accession No
		Locality	Texture	Growth (mm/d)	Color	Chlamydospore				Conidium				ITS
						Color	Shape	Length (μm)	Width (μm)	Color	Shape	Length (μm)	Width (μm)	
1	Hp1	Gansu	Villous	7.1	White	light brown	ellipsoid	6.5–25.1	4.5–15	Colorless	cylinder, rhabditiform	6.7–25.8	2.5–7.8	MK583555
2	Hp2	Gansu	Villous	7.6	White	-	-	-	-	-	-	-	-	MK584644
3	Hp3	Fujian	Villous	12.6	White	brown	Pear-Shaped	24–44	14–28	Colorless	cylinder, rhabditiform	5.9–17.7	2.9–8	MK584650
4	Hp4	Fujian	Villous	12.7	White	brown	Pear-Shaped	24–42	14–38	Colorless	cylinder, rhabditiform	9–30	3–11	MK583561
5	Hp5	Fujian	Villous	12	White	light brown	ellipsoid	13.6–30.6	7.8–19.4	Colorless	cylinder, rhabditiform	5.4–16.9	3.1–10.7	MK583565
6	Hp6	Shandong	Concentric circles	10.4	brown	brown	Pear-Shaped	24–48	15–30	Colorless	cylinder, rhabditiform	7–17	2–7.5	MK584645
7	Hp7	Fujian	Concentric circles	12	brown	brown	Pear-Shaped	18–42.5	13.7–27.7	Colorless	cylinder, rhabditiform	8–17.2	2.8–7.6	MK583576
8	Hp8	Shandong	Concentric circles	9.7	brown	-	-	-	-	-	-	-	-	MK584648
9	Hp9	Gansu	Villous	9.4	brown	brown	Pear-Shaped	18.4–35.4	12–27.7	-	-	-	-	MK584651
10	Hp10	Hubei	Concentric circles	12	brown	brown	Pear-Shaped	25.5–40.5	14–32	Colorless	cylinder, rhabditiform	8.3–24	2–8	MK584647
11	Hp11	Shandong	Concentric circles	12.7	brown	light brown	ellipsoid	17.2–35.7	10–30	Colorless	cylinder, rhabditiform	7.8–20	3–8.6	MK584649
12	Hp12	Shandong	Concentric circles	11.1	brown	brown	Pear-Shaped	26.2–41.6	19.8–28	Colorless	cylinder, rhabditiform	5.2–17.6	3.2–7.6	MK584646
13	Hp13	Fujian	Concentric circles	14	brown	brown	Pear-Shaped	20–36	10–25	Colorless	cylinder, rhabditiform	6.7–18	2.5–5.2	MK584655
14	Hp14	Fujian	Concentric circles	15	brown	light brown	ellipsoid	19–35.3	12–25	Colorless	cylinder, rhabditiform	6–18	4.6–6	MK584656
15	Hp15	Shandong	Concentric circles	12.5	brown	brown	Pear-Shaped	21–35	17.1–28	Colorless	cylinder, rhabditiform	8.8–20.7	3.1–12	MK584652
16	Hp16	Fujian	Concentric circles	11.6	brown	light brown	ellipsoid	9.9–43.5	5.5–31	Colorless	cylinder, rhabditiform	6.5–20	2.8–5.8	MK584653
17	Hp17	Fujian	Concentric circles	12.5	brown	brown	Pear-Shaped	25–42	18.8–30	Colorless	cylinder, rhabditiform	5–16	2–5.8	MK584654
18	Hp18	Fujian	Concentric circles	14.2	brown	brown	Pear-Shaped	23–41	23–27	Colorless	cylinder, rhabditiform	6.6–21.6	1.9–6.5	MK583447

**Table 2 pathogens-08-00179-t002:** Pathogenic variability of *Mycogone perniciosa* isolates on different *Agaricus bisporus* strains.

^1^*M. perniciosa* Isolates	^2^ Different *Agaricus bisporus* Strains					
	CCMJ1020	CCMJ1053	CCMJ1036	CCMJ1074	CCMJ1106	CCMJ1110
Hp1	S	R	R	R	R	R
Hp2	S	S	R	S	R	R
Hp3	S	S	S	S	R	R
Hp4	S	S	R	S	R	R
Hp5	S	S	R	R	R	R
Hp6	S	S	R	S	R	R
Hp7	S	R	R	R	R	R
Hp8	S	R	S	R	R	R
Hp9	S	R	R	R	R	R
Hp10	S	S	S	S	S	S
Hp11	S	R	S	R	R	R
Hp12	S	R	S	S	S	S
Hp13	S	S	R	S	R	R
Hp14	S	S	R	R	S	R
Hp15	S	R	S	R	R	R
Hp16	S	S	R	R	R	R
Hp17	S	S	R	R	S	R
Hp18	S	S	R	R	R	R

^1^*M. perniciosa* isolates with different pathogenic variability. ^2^
*A. bisporus* strains were classified as resistant (R) or susceptible (S) to *M. perniciosa* based on visual disease scores.

**Table 3 pathogens-08-00179-t003:** Primer pairs used in AFLP analysis.

No.		Primer	Sequence (5′–3′)	Application
1	F	HindIII 1	CTC GTA GAC TG CGT ACC	Adaptor
2	R	HindIII 2	AGC TGG TAC GCA GGT CTA C	
3	F	MseI 1	AGC TGG TAC GCA GGT CTA C	
4	R	MseI 2	TAC TCA GGA CTC AT	
5	F	HindIII	AGA CTG CGT ACC AGC TTA	Non-selective Pre-amplification
6	R	MseI	GAT GAG TCC TGA GTA AC	
**Hind III (5 ng/μL)**				
7	F	Hind III A	AGA CTG CGT ACC AGC TTA AC	PCR Selective amplification
8	R	Hind III B	AGA CTG CGT ACC AGC TTA AG	
9	F	Hind III C	AGA CTG CGT ACC AGC TTA CA	
10	R	Hind III D	AGA CTG CGT ACC AGC TTA CT	
11	F	Hind III E	AGA CTG CGT ACC AGC TTA CC	
12	R	Hind III F	AGA CTG CGT ACC AGC TTA CG	
13	F	Hind III G	AGA CTG CGT ACC AGC TTA GC	
14	R	Hind III H	AGA CTG CGT ACC AGC TTA GG	
**MseI primers (30 ng/μL)**				
15	F	FAM mark MseI-1	GAT GAG TCC TGA GTA ACA A	PCR Selective amplification
16	R	FAM mark MseI-2	GAT GAG TCC TGA GTA ACA C	
17	F	FAM mark MseI-3	GAT GAG TCC TGA GTA ACA G	
18	R	FAM mark MseI-4	GAT GAG TCC TGA GTA ACA T	
19	F	FAM mark MseI-5	GAT GAG TCC TGA GTA ACT A	
20	R	FAM mark MseI-6	GAT GAG TCC TGA GTA ACT C	
21	F	FAM mark MseI-7	GAT GAG TCC TGA GTA ACT G	
22	R	FAM mark MseI-8	GAT GAG TCC TGA GTA ACT T	

NB: F = forward primer; R = reverse primer.

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
