# Peer review of "Genetic and Pathogenic Variability of Mycogone perniciosa Isolates Causing Wet Bubble Disease on Agaricus bisporus in China"

_pathogens, 2019, doi:10.3390/pathogens8040179_

Round 1

Reviewer 1 Report

Dear Authors,

the article "Genetic and Pathogenic Variability of Hypomyces perniciosus Isolates Causing Wet Bubble Disease on Agaricus bisporus in China" covers a good approach to the understanding of the Chinese genetic diversity of the causative agent of wet bubble disease in mushrooms (the anamorph synonym Mycogone perniciosa). It could be considered for publication in Pathogens.

There is a good number of isolates collected from different locations in China, they have been identified and described separately by two independent techniques. morphology (but some mistakes have been detected in this description, including the name given to the parasite, since the authors used the teleomorph, sexual stage, Hypomyces perniciosus, but the specimen described is clearly the anamorph, asexual stage, Mycogone perniciosa) and molecular. in addition, they present a genetic characterisation based on AFLP that can be useful to understand mechanisms of infection.

However, I have detected a number of issues that require a major revision of the article.

Please find attached my comments.

Best regards.

Author Response

Response to Reviewer 1:

Thank you for your comments. We have improved the overall English language of the manuscript. Our answers to your points are as follows.

Point 1: Line 2-4: In the title and subsequently along with the text the authors should use Mycogone perniciosa as the name of the parasite. This is because Hypomyces perniciosus is the teleomorph synonym, the sexual reproductive stage for this species, morphologically ascospores and ascus are representative structures of this stage. However, Mycogone perniciosa is the anamorph of the species, the asexual stage of the fungus, characterized for the presence for conidia or chlamydospores. in summary, the wet bubble disease is caused by the asexual stage: Mycogone perniciosa.

Response 1: Thank you. As suggested, we have changed the name Hypomyces perniciosus to Mycogone perniciosa in the title and subsequently along the text.

Point 2: Line 25: Rephrase the sentence for better understanding: "Through knowledge of genetic diversity, phenotypic..."??

Response 2: Thank you. As suggested, we have rephrased the sentence to “Through knowledge of the genetic diversity, phenotypic….”

Point 3: Line 76: Mycogone perniciosa produces two different kind of conidia, rod-shaped  bacilliform conidia called phialoconidia or phialospores, and bulb light aleurispores (shown in Figure 1). Please re-chack taxonomical features, both structures are sucestible of create an outbreak, so it is important to discriminate and characterise (Glamočlija et al., 2008. Journal of microscopy, 232(3), 489-492; Holland & Cooke, 1990. Mycological Research, 94(6), 789-792). 

Response 3: Thank you. The spore of M. perniciosa in the anamorph stage are divided into two types: one is the conidia with thin cell wall, ellipse, colorless and transparent, which are born on the conidia pedicel with a diameter of 9.77×4.25µm-12.77×6.05µm(averge,n=60) and the other one is the chlamydospore composed of the upper and lower cells. The upper cell is yellow-brown to gray-brown, and the lower cell is colorless and transparent, its size is from 11.36×8.08µm -33.43×25.94µm(averge,n=60).

Point 4: Figure 1: the quality of pictures C and F can be significantly improved. Scale bar is missing in picture C. Picture C and F shows a messy cluster of coloured aleurispores and some hyphae, probably some arrows pointing the detailed structures will help to understand taxonomical features.

Response 4: Thank you. As suggested, we have provided quality pictures for figure 1 C and F. The scale bar as well as red arrows pointing to the for taxonomical features.

Point 5: Figure 2. Quality of pictures is low, A and C show blurring and lack of clarity in the photos, good pictures are required when showing symptoms of disease.

Response 5: Thank you. As suggested, we have provided good quality pictures for figure 2 A and C.

Point 6: Line 170. from my point of view the fact that white strains show less pathogenicity is significant, I wonder if this is because conidia are colorless, the sporulation degree is lower than in yellow/brown colonies?? I think this point needs a rigorous discussion since can be related to the production of pigments in other mushroom mycoparasites such as Cladobotryum mycophilum (Carrasco et al., 2017. Spanish Journal of agricultural research, 15(2), 19). Please make an effort on this point since even your technical hypothesis could be of high value for other researchers.

Response 6: Thank you. As suggested, we have laid more emphasis on the pathogenicity of the white isolates. It reads “The white isolates of M. perniciosa showed less pathogenicity compared to the yellow/brown isolates. This may be due to the colorless conidia and a lower degree of sporulation compared to the yellow/brown colonies. For example, Fletcher et al. [4] reported that non-pigmented white isolates of  M. perniciosa are less pathogenic than the pigmented isolates. ”  

Point 7: Line 192: It should be useful for the understanding of these results to show AFLP banding patterns produced for representative strains from detected clusters. Basically, the gels obtained.

Response 7: Thank you. As suggested, we have provided the gels for Hp11 and Hp 12 as figure 4.

Point 8: Line 248: Specify growing conditions, fruiting requires low CO2 concentration, high values are related to the germination of the mycelium in compost and casing, to induce fruiting ventilation is required. Therefore, when talking of fruiting 1200-1500 ppm should be the optimal pressure.

Response 8: Thank you. As suggested, we have changed the CO2 concentration to 1200-1500 ppm

Point 9: Line 266-271. I do not fully understand the symptoms you have considered. For instance, are minor symptoms related to cap spotting? Are the symptoms considered reflected to masses of featureless or distorted Agaricus tissue?

Is this based on the number of basidiomes showing disease against the total mushrooms harvested from the plots?

Please this needs clarification in order to the scientific community to understand and if required, reproduce the trial.

Response 9: Thank you. The disease scale is based on the number of basidiomes showing disease against the total mushrooms harvested from the basket.

Point 10: Line 290: Please check is these conditions are accurate: "...incubation at 37ºC for 5h, 8ºC???

Response 10: Thank you. The sentence has been changed to “The reaction mixture was centrifuged for 15 s, incubated at 37 for 5 h, held at 8 for 4 h and stored overnight at 4 .”

Point 11: The authors should make an effort to update references. 45% of the cited ref. is more than 10 years old, that should not occur in more than 10% in order to build a discussion in the present. Please check carefully and refurbish this section correspondingly.

Response 11: Thank you. As suggested, we updated the references. 37.5% of the cited references are more than 10 years old. However, most of these references are found in the materials and methods 

Reviewer 2 Report

This is a well set out and executed piece of research. It contributes to the knowledge of this disease by identifying variation in morphology and virulence on a set of isolates from China.

It is particularly useful the detection of white colony isolates which are hypovirulent. This phenomenon has already been observed on other fungal pathogens and has been associated to mycoviruses. Mycoviruses are highly relevant in order to set up biocontrol methods (Eg. Control of Cryphonectria parasitica causing chestnut blight). However, in this work, even though authors formulate several hypothesis to explain these different levels of virulence, they did not confirm any of them. Such studies should have complemented this research.

Other comments:

Please, check discrepancy regarding the information given in the text (lines 78-79) and Table 1. Chlamydospores and/or conidia not observed on Hp isolates.

On line 79: any sort of quantification of conidia production would have been helpful to assess differences between isolates with white or brown colony.

Figures 1 and 2: Please, check captions (error in letters).

Part 2.2 missing (line 105).

Too many references cited, whose number could easily be reduced.

Author Response

Response to Reviewer 2:

Thank you for your comments. We have improved the overall English language of the manuscript. Our answers to your points are as follows.

Point 1: Please, check discrepancy regarding the information given in the text (lines 78-79) and Table 1. Chlamydospores and/or conidia not observed on Hp isolates.

Response 1: Thank you. As suggested, we have checked the discrepancy and corrected the error. It now reads: “On PDA chlamydospores were not observed on Hp2 and Hp9, similarly, conidia were not observed for H2, Hp8, and Hp9 (Supplementary Table 1).”

Point 2: On line 79: any sort of quantification of conidia production would have been helpful to assess differences between isolates with a white or brown colony.

Response 2: As suggested, we have quantified the conidia production and it is presented in Supplementary Table 1

Point 3: Figures 1 and 2: Please, check captions (error in letters).

Response 3: Thank you. We have checked the captions and corrected errors accordingly. The caption now reads:

“Figure 1. Colony morphology and microscopic characteristics of M. perniciosa:  Colony color A: upper, B: under  C: microstructure for isolate HP10. Colony color D: upper, E: under, F: microstructure for isolate Hp2. aaleuriospore, b: hypha.”

 “Figure 2. Pathogenicity tests for M. perniciosa on A. bisporus strain 14:  Isolate Hp10 A: Diseased basidiome occurring in the field, B: Control, C: wet bubble disease from pathogenicity test. Isolate Hp2 D: Diseased basidiome occurring in the field, E: Control, F: wet bubble disease from pathogenicity test”

Point 4: Part 2.2 missing (line 105).

Response 4: Thanked you. We have corrected the headings. It reads as follows: “2.1. Morphological characterization and phylogenetic analyses of Mycogone perniciosa , “2.2. Pathogenicity variability ”, and “2.3. AFLP Analysis

Point 5: Too many references cited, whose number could easily be reduced.

Response 5: Thanked you. As suggested, we have reduced the reference cited to 32.  

Round 2

Reviewer 1 Report

In my opinion the quality of the article has improved and can now be considered for publication in Pathogens.